# Extracellular Vesicles’ Role in the Pathophysiology and as Biomarkers in Cystic Fibrosis and COPD

**DOI:** 10.3390/ijms24010228

**Published:** 2022-12-23

**Authors:** Sante Di Gioia, Valeria Daniello, Massimo Conese

**Affiliations:** Department of Clinical and Experimental Medicine, University of Foggia, 71122 Foggia, Italy

**Keywords:** extracellular vesicles, apoptotic bodies, microvesicles, exosomes, miRNAs, cystic fibrosis, chronic obstructive pulmonary disease, pathomechanisms

## Abstract

In keeping with the extraordinary interest and advancement of extracellular vesicles (EVs) in pathogenesis and diagnosis fields, we herein present an update to the knowledge about their role in cystic fibrosis (CF) and chronic obstructive pulmonary disease (COPD). Although CF and COPD stem from a different origin, one genetic and the other acquired, they share a similar pathophysiology, being the CF transmembrane conductance regulator (CFTR) protein implied in both disorders. Various subsets of EVs, comprised mainly of microvesicles (MVs) and exosomes (EXOs), are secreted by various cell types that are either resident or attracted in the airways during the onset and progression of CF and COPD lung disease, representing a vehicle for metabolites, proteins and RNAs (especially microRNAs), that in turn lead to events as such neutrophil influx, the overwhelming of proteases (elastase, metalloproteases), oxidative stress, myofibroblast activation and collagen deposition. Eventually, all of these pathomechanisms lead to chronic inflammation, mucus overproduction, remodeling of the airways, and fibrosis, thus operating a complex interplay among cells and tissues. The detection of MVs and EXOs in blood and biological fluids coming from the airways (bronchoalveolar lavage fluid and sputum) allows the consideration of EVs and their cargoes as promising biomarkers for CF and COPD, although clinical expectations have yet to be fulfilled.

## 1. Introduction

### 1.1. General Role of EVs in Various Pathological Processes

Cells secrete or shed small vesicles from their membrane called extracellular vesicles (EVs), functioning as a long distance cell-to-cell communication mechanism [1]. EVs mediate the transport of several molecules that can exert positive or detrimental effects on their target cells. Indeed, via both autocrine and paracrine signaling, EVs can induce many changes in the biological activity of cells, such as the activation of the inflammatory process, the increase in cell proliferation, etc. In recent years, the role of EVs in several pathological conditions has been clarified, and it is well known that their specific cargo is associated with different pathophysiological statuses. Hereafter, we give a brief overview of EVs’ roles in some pathological conditions. In the cardiovascular system, EVs can be secreted by different cell types (e.g., cardiomyocytes, smooth muscle cells, macrophages, monocytes, fibroblasts, and endothelial cells), and can trigger several pathological processes [2]. It must be said that the *primum movens* in the pathogenesis of many cardiovascular diseases is represented by the endothelial dysfunction associated with atherosclerosis. EVs secreted by various cells, such as leukocytes, platelets, smooth muscle cells and endothelial cells, can activate the endothelium, promoting monocyte adhesion and transmigration, all of which are fundamental processes in inducing atherosclerosis [3]. In particular, platelet-derived EVs are involved in several physiological processes such as endothelial cell proliferation, survival, migration, and vessel formation. Platelet EVs promote inflammation via their cargo, which contain mediators capable of activating the endothelium and facilitating cell–cell interaction. Some studies have demonstrated that platelet-derived EVs are able of promoting tissue repair after cardiac damage [4].

Liver is another organ in which the pathophysiological role of EVs has been clarified. EVs secreted by hepatocytes, Kupffer cells, cholangiocytes, hepatic stellate cells and liver sinusoidal endothelial cells, and they exert an important role in many process such as liver injury, regeneration, fibrosis, inflammation, ductular reaction, and cancer development [5].

Emerging studies suggest a role for EVs both in neuroinflammation and neurodegeneration. EVs have both beneficial and detrimental roles in neuroinflammation [3]. Indeed, EVs act as “shuttles” of pro-inflammatory mediators, facilitating their transport from damaged cells to neighboring naïve neural cells [6]. Microglial cells, which are the first to respond during neuroinflammation, once activated can proliferate and in turn release EVs, allowing the spreading of danger signals across the brain [7]. In neurodegenerative diseases, such as Alzheimer’s disease (AD) and Parkinson’s diseases (PD), EVs work as carriers of pathogenic proteins. Indeed, in the case of AD and PD, EVs allow the spreading of the causative agents: αβ peptide and phosphorylated tau and α-Syn, respectively.

In cancer research, many studies have shown that EVs contribute to tumor progression by promoting metastatic dissemination and resistance to chemotherapy [8]. EVs from tumor cells are involved in the crosstalk between stromal cells and cancer cells as well as in reprogramming cancer cells, promoting their epithelial–mesenchymal transition. Moreover, EVs can encapsulate microRNAs (miRNAs) and proteins, which can transform nonmalignant cells or improve chemotherapy resistance. EVs can also alter the expression of molecules involved in the immune response against cancer cells. All of these studies demonstrated that EVs can “educate” cancer cells by means of their cargo, in particular proteins and miRNAs.

Finally, EVs have an important role in infectious diseases where the crosstalk between host cells and pathogenic organisms is fundamental [8]. In vitro, it has been demonstrated that EVs, derived from hepatoma cells infected with HCV, promote infection without interaction between target cells and viruses. Moreover, there is an increasing interest in studying EVs as a marker of infection. Interestingly, studies concerning HIV-1-infected patients have demonstrated that the quantity of EVs is informative of the activity of viral infection. Again, the modification of the EVs’ cargo, in terms of microRNAs, can be a molecular signature of response to therapy.

In respiratory medicine, a large body of studies has demonstrated the involvement of EVs in the pathogenesis of lung diseases such as lung cancer, pulmonary fibrosis and SARS-CoV-2 infection. Indeed, EVs’ cargo can vary in terms of the dependence of the different pathophysiological status of the disease [9]. In lung cancer, EVs secreted by malignant cells have important roles in oncogenesis and tumor progression [10]. Lung cancer cells’ EVs are involved in the stimulation of angiogenesis, enhancing the expression of vascular endothelial growth factor (VEGF) [11]. Moreover, cancer cells’ EVs can modify the “immunological microenvironment” of the tumor. Indeed, EVs can induce mesenchymal stem cells (MSCs) to acquire a pro-inflammatory phenotype, or induce a tolerogenic response to attenuate the killing activity of CD8 T cells, both of which are fundamental processes in tumor growth [12,13]. In addition, cancer cells’ EVs can promote the epithelial-to-mesenchymal transition [14] as well as the pre-metastatic niche [15]. In idiopathic pulmonary fibrosis (IPF), EVs have shown potential as biomarkers [9]. In particular, using a murine model of lung fibrosis, the role of an EV–miRNA as a potential marker of different stages of IPF was clarified [16]. Interestingly, the same miRNA was increased in the serum of IPF patients. As compared to conventional biomarkers of IPF, such as Krebs von den Lungen-6 (KL-6) and matrix metalloproteases, EVs and miRNAs seem to be quite specific to cell conditions in pulmonary fibrosis [9]. Recently, many studies were carried out to evaluate the role of EVs in SARS-CoV-2 infection [17]. It was demonstrated that several cytotypes represent the first source of EVs [18,19,20]. Interestingly, the presence of ACE2 (receptor angiotensin-converting enzyme 2) in EVs released by some types of cells [21] suggests an important role of these vesicles in promoting infection, given that ACE2 is fundamental for the fusion between SARS-CoV-2 viral particles and the host cell membrane. During respiratory infections by coronaviruses, the serum level of EVs increases [22,23]. Moreover, EVs may also activate host immune responses if they carry viral and self-antigens [24]. The systemic increase in exosomes reported and detected during SARS-CoV-2 infection [25,26] may be closely related to the pathological events characteristic of COVID-19. Indeed, it is well known that SARS-CoV-2 infection is associated with an increased risk of thromboembolic events. In COVID-19 patients, the contribution of platelet EVs to thrombotic events is mainly due to their higher expression of procoagulant proteins, such as tissue factor [4].

### 1.2. CF and COPD

EVs also contribute to the pathogenesis of chronic inflammatory respiratory diseases, such as asthma, cystic fibrosis (CF) and chronic obstructive pulmonary disease (COPD) [27]. CF and COPD share a lung pathology whose hallmarks are mucus overproduction, opportunistic bacterial infections and a chronic inflammatory response, eventually leading to bronchiectasis and lung failure. While CF is caused by mutations in the CF transmembrane conductance regulator gene (CFTR), COPD is considered an acquired CFTR dysfunction, mostly due to cigarette smoking. Indeed, these two diseases display a substantial overlap in clinical phenotype, and also more recently in the role of EVs in their pathophysiological processes. The aim of this review is to focus on the evidence of EVs’ role in the pathomechanisms of CF and COPD, as a novel portrait of these diseases that could also advance biomarker options.

#### 1.2.1. CF

CF is an autosomal recessive condition that affects various organs and tissues, and is the lung disease responsible for the majority of morbidity and mortality cases. The CFTR protein regulates the exchange of chloride, bicarbonate and sodium ions through epithelial membranes. More than 2000 variations have been recognized in the CFTR gene, among which only a limited subset is known to be disease-driven [28]. CFTR mutations have been classified into six classes, depending on the effect on CFTR protein maturation and transport to the plasma membrane, channel opening, gating and recycling [29]. The most frequent mutation, affecting ~80% of the Caucasian population, where CF shows the highest incidence, is the deletion of phenylalanine at position 508, i.e., *F508del*, causing misfolding, an altered interaction with chaperonins at the endoplasmic reticulum level, and premature proteasome-mediated degradation [30,31]. However, this class II mutation also causes defects belonging to class III (reduced channel gating) [32,33] and class VI (altered recycling) [34,35,36] mutations.

In CF, CFTR loss/malfunction causes a disrupted ion and fluid flux through the airway epithelium resulting in the production of thick/viscous mucus, which is due to the pathological increment in proteins, mucin and biological polymers [37]. Increased airway mucus viscosity, resulting in the mucopurulent obstruction of small and medium-size bronchioles and bronchiectasis [38], promotes bacterial infection and inflammation, which may proceed until patients die of respiratory insufficiency [39]. A neutrophil-dominated peribronchial and endobronchial inflammation is the hallmark of CF lung inflammatory disease, although the propagation of inflammation and damage to the lungs is not well known [40]. Although neutrophils are considered the major immune cell type responsible for inflammation and tissue damage in CF lungs [41], other innate immune cells have been found to be dysregulated in CF, such as epithelial cells, macrophages, and dendritic cells [42,43,44]. However, the interplay among these cell types is scarcely known. Nowadays, the clinical history of CF has been profoundly changed by the introduction to the therapeutic toolbox of small-molecule drugs modifying the expression and function of mutated CFTRs. To keep with the *F508del* mutation and its multiple alterations, a cocktail of correctors (inciting correct trafficking to the plasma membrane) and potentiators (increasing channel-opening probability) are assumed to control CF disease in all patients bearing at least *F508del* on at least one allele, representing nearly 90% of CF individuals [45].

#### 1.2.2. COPD

COPD is characterized by a spectrum of conditions, including small-airway inflammation with subsequent narrowing and emphysema, which causes irreversible damage to distal parts of the lung [46,47]. Cigarette smoke (CS) is the main cause of these pathologic alterations, resulting in inflammation and remodeling through the modification of bronchial epithelial cells, the differentiation of fibroblasts into myofibroblasts [48], and eventually the EVs produced by these cells [49]. Inflammation, oxidative stress, and protease–antiprotease disproportion are the hallmarks of the pathogenesis of chronic bronchitis, small-airway disease and emphysema [50].

COPD and CF have in common many clinical and pathologic features, such as the airflow limitation, neutrophilic inflammation and the enhancement of lung mucus viscosity [51,52]. Of the two classic COPD phenotypes, prevalent emphysema and prevalent chronic bronchitis (CB) [53,54], the latter exhibits pathologic features similar to CF, including mucin overproduction and mucus accumulation. The CB phenotype affects nearly two-thirds of COPD patients [55]. While CF is a genetic disease due to CFTR mutations, COPD is typically the result of smoke and the inhalation of toxic substances. A deficiency of α1-antitrypsin (α1AT), as a cause of COPD, is generally considered to be rare, and less than 3% of people with COPD have an α1AT deficiency [48]. Individuals with an α1AT deficiency are at a high risk of COPD, especially if they are smokers, due to unopposed elastase activation [56,57], as well as excessive apoptosis [58,59] and lung inflammation [60,61,62].

Recent in vitro and in vivo findings have determined that COPD is considered an obstructive lung disease due to acquired deficiency of CFTR. Mechanistically, CS-induced oxidative stress and increased cytosolic Ca^2+^ levels as well as CS toxic components, which are considered the major causes of COPD, can lead to reduced CFTR levels in airway epithelia [63]. CFTR loss/dysfunction is due to multiple mechanisms, including reduced CFTR transcription, diminished CFTR protein by heightened protein degradation, and an increase in its internalization rate followed by the retrograde trafficking of CFTR to the endoplasmic reticulum [64,65,66,67]. Among the toxic CS constituents, acrolein, cadmium and manganese are thought to cause CFTR dysfunctions in vitro and in vivo [68,69,70]. Moreover, CS could also alter CFTR lipid rafts in macrophages, impairing bacterial phagocytosis and killing activity [71]. Overall, these findings suggest that, as in CF, the first steps in COPD pathogenesis (especially regarding CB) are represented by impaired ion and liquid homeostasis, mucociliary clearance reduction, and the ensuing opportunistic infection and inflammation [63]. However, the COPD pathogenesis of early as well of chronic disease is not well known, leaving room for other pathomechanisms to be found and studied. The similarity of COPD to CF extends so far that CFTR modulators are being evaluated for CFTR rescue and clinical efficacy in COPD patients [72,73,74].

## 2. Extracellular Vesicles (EVs) and Lung Diseases

### 2.1. Biogenesis of EVs

Most prokaryotes and eukaryotic cells release EVs into the extracellular environment [75,76,77]. EVs consist of a phospholipid bilayer and have a role in cell-to-cell communication and disease pathogenesis [78,79]. As reported in the Minimal Information for Studies of Extracellular Vesicles (MISEV) guidelines, published by the International Society for Extracellular Vesicles (ISEV), depending on their size, density, composition (surface markers), biogenesis and secretory mechanisms [80], EVs can be distinguished into (1) apoptotic bodies, (2) microvesicles and (3) exosomes. Among these three structures, the apoptotic bodies are those with the largest size, in a range between 1000 and 5000 nm. Apoptotic bodies (ABs) derive from the fragmentation of the cell in the process of apoptosis and are released by budding from the plasma membrane. Apoptosis-derived EVs (ApoBDs) may contain cellular organelles, membranes, and cytosolic contents, but also noncoding RNAs and DNA fragments [79]. The density of ApoBDs is in the range between 1.18 and 1.28 g/mL [81]. Microvesicles (MVs), also known as microparticles (MPs) or ectosomes, are smaller than apoptotic bodies, ranging from 150–1000 nm. MVs originate from the outward budding of the plasma membrane and are subsequently released extracellularly [82]. These EVs are enriched with lipids, particularly phosphatidylserine (PS) [83,84]. The sucrose gradient density of MVs, lower than ApoBDs, is between 1.25 and 1.30 g/mL [85,86,87]. Exosomes (EXOs) are characterized by a double phospholipid layer and a diameter between 30 and 150 nm. The biogenesis of exosomes is a complex process that depends on mother-cell-stimulating signals [88] and begins with the formation of early endosomes [89,90,91]. The early sorting endosomes become late sorting endosomes [92] with the formation of intraluminal vesicles (ILVs) within the lumen of the endosome. Multivesicular bodies (MVBs) are formed thanks to ILVs which, during their development, randomly incorporate portions of the cytosol, as well as transmembrane and peripheral proteins, into the invaginating membrane [87]. Some MVBs are degraded in lysosomes by the endosomal sorting complex required for transports (ESCRTs)-dependent mechanism, while a good portion of these MVBs fuses with the plasma membrane and secretes exosomes via exocytosis into the extracellular space, using RAB GTPases (ESCRT-independent mechanism) [93,94].

The ESCRT mechanism, which is composed of the various ESCRT complexes (ESCRT-0, -I, -II, -III) and some proteins such as vacuolar protein sorting-associated protein 4 (Vps4), apoptosis-linked gene 2-interacting protein X (ALIX) and tumor susceptibility gene 101 protein (TSG101) [90,95], takes part in the process of the biogenesis/degradation of MVBs in a ubiquitination-dependent manner [96]. The ESCRT-0 complex is recruited to the endosomal membrane, while ESCRT-I and -II are required to cause the deformation of the membrane, allowing the sorting of cargo molecules in the MVBs. Subsequently, the ESCRT-III/Vps4 complex regulates the concentration of the components of the MVBs’ cargo and separates the vesicles from the plasma membrane [97,98].

Although the three main classes of EVs have been identified, as previously mentioned, as EXOs, MVs and ApoBDs, the issue of EV diversity has not been completely elucidated and is the subject of intense investigation by ISEV and independent researchers. Currently, it can be said that cells release a spectrum of heterogeneous populations of EVs with overlapping sizes [99]. Simply, it has been observed that EV subtypes can be differentiated according to: (1) physical characteristics, such as size or density; (2) biochemical composition; and (3) descriptions of conditions or cell of origin [100,101]. In terms of size, we can define small EVs (S-EVs) as being <200 nm in diameter and large and/or medium EVs (m/l-EVs or L-EVs) as over 200 nm [80]. As for the molecular content, these EVs differ since L-EVs have more DNA, CD9, and Annexin A1, while S-EVs carry CD63 and CD81 [100]. Moreover, it has been proposed that, concerning miRNAs, there exist different EV subpopulations with unique characteristics and miRNA contents [99]. Finally, other EV subpopulations, called large oncosomes and originating directly from the plasma membrane, are released specifically by cancer cells, and they are larger than any other EV, i.e., between 1 and 10 μm [102].

Besides the inherent heterogeneity of EVs, complexity also concerns EXOs, which can include different subpopulations based on their size, content (cargo), functional impact on recipient cells, and cell of origin [103]. Three types of EXOs have been identified according to their size (40–75 nm, 75–100 nm, or 100–160 nm), their biomarker (tetraspanins CD63, CD9, or CD81) and their functional heterogeneity. In this last case, one EXO subtype can induce cell survival, while another set induces apoptosis, and a different set induces immunomodulation, etc., in different target cell types [103,104]. To illustrate this complex diversity, Zhang et al. [105], by using asymmetric flow field-flow fractionation, identified two exosome subpopulations (large exosome vesicles, Exo-L, 90–120 nm; small exosome vesicles, Exo-S, 60–80 nm) and discovered an abundant population of nonmembranous nanoparticles termed “exomeres” (~35 nm), and each of these subpopulations were associated with specific phenotypes and cargoes. The heterogeneity in terms of both the size and composition of EXOs is not surprising given the different protein complexes involved in ESCRT machinery and the existence of the ESCRT-independent pathway of EXO biogenesis.

The assessment of EV heterogeneity is the focus of several methods that capture EV populations and subpopulations, including ultrafiltration, differential ultracentrifugation, gradient ultracentrifugation, precipitation, size-exclusion chromatography, immune-affinity capture, mass spectrometric immunoassay, etc. [80,99,106,107,108]. An initial characterization of EV subpopulations, obtained after isolation, can be performed morphologically by scanning electron microscopy (SEM) [109], transmission electron microscopy (TEM) [110,111,112] or atomic force microscopy (AFM) [77,80,106,108].

Since EXOs are the EV subpopulation characterized by a distinct intracellular regulatory process that likely affects their composition, and possibly their function, when secreted into the extracellular space and interacting with recipient cells [113,114,115], this review is intended to place particular emphasis on EXOs and to discuss other EVs when a thoughtful comparison is useful.

### 2.2. EVs’ Cargo

The Bradford or micro-bicinchoninic acid (BCA) protein assay is used to quantify the vesicular proteins that are subsequently separated and specifically detected by Western blot analysis using different monoclonal antibodies, i.e., CD9, CD63, CD81, Hsp70 [109], MHC II, CD40, CD80, CD86, and CD54 [77,80,106,108].

Purified EVs can thus undergo characterization with respect to their cargoes, such as metabolites and lipids [116,117,118], proteins [117,119], transcripts (prevalently mRNAs, miRNAs and long noncoding (lnc) RNAs, but also small nuclear RNA, small nucleolar RNA, noncoding RNA, long intergenic noncoding RNA, piwi-interacting RNA) [117,120,121], and genetic material (single-strand DNA, double-strand DNA, mitochondrial DNA) [117,122]. The vesicles’ cargo depends on the source from which the EVs originate and on their biological state (e.g., transformed, differentiated, stimulated, and stressed) [89,123]. Furthermore, based on the molecular content of EVs, it is possible to understand the physiological and pathological state of the cells [124,125,126].

Exosomes consist of a lipophilic shell with lipids and intramembrane ligands and receptors [127]. These vesicles are detected in biological fluids such as blood [128], sputum [129], cerebrospinal fluid [130], bronchial alveolar lavage fluid (BALF) [131], and urine [132], but they can also be secreted by various cell types such as bronchial epithelial cells [133], mesenchymal stem cells [134], macrophages [135], dendritic cells [136], natural killer (NK) cells [137], T cells [138], and B cells [139]. Exosomes are released into the extracellular environment and directed to different organs such as the lung, liver, kidney, pancreas, spleen, gastrointestinal tract, etc. [140,141,142]. Thanks to EXOs’ amphiphilic properties, hydrophobic and hydrophilic bioactive molecules can be incorporated into them [143].

EXOs have a very complex composition. They are characterized by a high and specific protein content (~4563 proteins) [144,145] derived from the plasma membrane, the cytosol, and the endocytosis pathway. Particularly, EXOs are rich in the most common proteins CD9, CD81, CD82, CD63, CD106 of the tetraspanin family [89,103,146], which act as specific membrane markers [147] and play an important role in cell penetration, invasion and fusion; the heat shock proteins HSP60, HSP70, HSP90, which allow the peptide to be loaded onto MHC I; Alix, Flotillin-1 and TSG101 proteins [89,103,146], which are necessary during the biogenesis of exosomes and useful for the trafficking and release through the MVB formation [103]. Among the various proteins that are part of the exosomes’ cargo there are those located on the surface of the vesicles that can induce intracellular signaling by interacting with the target cells’ receptors. Furthermore, EXOs carry other proteins such as annexins (I, II, V e VI) and Ras proteins that promote the docking of vesicles and membrane transport and fusion [145], cell-specific class II proteins of the myosin heavy chain, which are incorporated only in exosomes isolated from APC cells [148], GTPases (EEF1A1, eEF2) and cytoskeletal proteins (actin, sinenin and moesin), enzymes and signal transduction proteins as well as lipids (approximately 194) [103,145,149], and functionally active genetic material including DNA, messenger RNA (mRNA), microRNAs (miRNAs), noncoding RNA (tRNAs, rRNAs), and metabolites [150,151].

miRNAs are small noncoding RNAs ranging in size from 20 to 22 nucleotides and are the most abundant component of the exosome’s cargo. Their internalization in exosomes is regulated by the presence of RNA-binding proteins via specific conserved miRNA motifs. For example, there are various RNA-binding proteins, such as SUMO protein (hnRNBA1), that can recognize GAGAG motifs of miRNA [152], or the synaptotagmin-binding cytoplasmic RNA-interacting protein that identifies the GGCU motif [153]. Previously, several studies have shown the presence of large amounts of extracellular miRNAs outside EXOs and MVs, which are often associated with Argonaute (Ago) proteins. These miRNAs could result from the death of many cells, remaining stable in the extracellular environment thanks to binding with the Ago-2 protein [154]. miRNAs, specifically those incorporated in exosomes, are actively secreted and thus involved in unidirectional intracellular trafficking, causing phenotypic changes of the recipient cells [155]. EXO miRNAs can regulate gene expression at the translational and post-translational levels [156]. All body fluids are rich in exosomes and therefore in the miRNAs packed inside the vesicles. For this reason, exosomes are utilized as noninvasive biomarkers [157], especially in cancer prognosis. For example, EXO miR-451a, miR-21 and miR-4257 are altered in lung cancer and thus associated with tumor progression and poor prognosis [158].

The intercellular signaling and the structural stability and rigidity of exosomes is normally guaranteed by several components such as cholesterol, arachidonic acid, diglycerides, sphingolipids (i.e., sphingomyelin and ceramide), phospholipids, glycerophospholipids including PS, phosphatidylcholine (PC), phosphatidylinositol (PI), and phosphatidylethanolamine (PE), but also bioactive lipids such as prostaglandins and leukotrienes [159]. Exosomes have a lipid content comparable to that present in the parental cell [160], but sometimes even higher for specific lipids (PS, PC, PI, cholesterol). This latter aspect increases the rigidity of the exosome membrane.

There are not many studies on the molecular composition of MVs, but the presence of lipids, proteins and nucleic acids similar to those of exosomes has been identified. The proteomic cargo of MVs is characterized by various molecules, already mentioned in EXOs, including Alix, TSG101, the tetraspanins CD9, CD81, CD63, CD86, CD40, integrins, selectins, and glycoproteins, e.g., GPIb [85,161], but also matrix metalloproteinases (MMPs) [85,162] and cytoskeletal proteins, e.g., β-actin [163]. From a lipidomic point of view, MVs’ membranes are enriched with different lipids, mostly PC (59.2%), but also sphingomyelin (SM), PE, PS, cholesterol and diacylglycerol [164,165,166,167,168,169]. Instead, ApoBDs, based on their origin, are composed of a myriad of proteins such as heat shock proteins, e.g., HSPB6, RAB11A, Annexins, e.g., ANXA6 or ANXA5, histones and cytosolic proteins [168,170]. Moreover, these vesicles are rich in PS [171]. In addition to exosomes, both MVs and ApoBDs are also characterized by the presence of nucleic acids including DNA, mRNA, miRNA and rRNA [172,173].

### 2.3. EVs in Lung Homeostasis

Epithelia are characterized by a higher turnover rate in comparison to other tissue compartments, as old cells are continually replaced by new cells derived from differentiation into the staminal niche. However, as compared to other organs (e.g., skin, gut), in physiological conditions, lungs are characterized by a very slow cellular turnover [174]. It is well known that lungs are daily exposed to noxious stimuli such as environmental pollutants and microorganisms, which can give rise to an inflammatory process. When damaged, lungs are able to regenerate and repair its different tissues. In the “steady state”, the cell renewal is very low, while after injury a regenerative response takes place thanks to interactions between stem cells and the surrounding environment [175]. Lungs are characterized by regenerative processes that respond to a kind of “regionalization”. Indeed, in the proximal airways, the progenitor compartment is represented by basal cells that self-renew and, in the presence of tissue injury, can differentiate into different cytotypes such as secretory, goblet, ciliated and neuroendocrine cells so as to guarantee the integrity of airway structures. Apart from basal cells, secretory cells possess self-renewing properties and a good differentiative activity [176,177,178]. Alveoli, which are representative of the distal region, are composed of alveolar type 1 (AT1) and alveolar type 1 (AT2), with the latter having a staminal/progenitor behavior, which is particularly important both during the formation of alveoli and after cell damage in adult lungs [179,180,181]. In addition, during lung regeneration, the stromal components supporting the epithelial staminal niche, e.g., mesenchymal and immune cells, seems to be very important [182].

In recent years, it has been demonstrated that EVs regulate the lungs’ homeostasis as well as their functions [183,184,185,186]. Many types of respiratory cells can release EVs (MVs and EXOs), including alveolar epithelial cells, alveolar macrophages, pulmonary vascular endothelial cells, airway and vascular smooth muscle cells, fibroblasts, stromal cells, and immune cells [93]. Recently, it has been demonstrated that EVs derived from airway epithelial cells are fundamental for both the communication among different epithelial cells and the homeostasis of airways and alveoli. When airway epithelial cells take up EVs derived from different airway cell populations, many molecules with important roles in cell life, such as proteins and miRNAs, are differently expressed or regulated. Recently, Kadota et al. [187] published a paper concerning the role EVs in the modulation of cell signaling involved in IPF, a disease in which the lung parenchyma is deeply compromised, resulting in poor patient prognosis. When given intratracheally in a mouse model of bleomycin-induced lung fibrosis, bronchial epithelial cell-derived EVs can attenuate WNT signaling, thereby suppressing the differentiation of myofibroblasts and epithelial cell senescence. The antifibrotic activity of bronchial epithelial cell EVs is higher as compared to that of mesenchymal stem cell EVs. It is interesting to point out that this effect is partly due to the miRNA cargo, which is able to negatively regulate the WNT–TGF-β crosstalk. In addition to the attenuation of the profibrotic cell phenotype and experimental lung fibrosis in vivo, they observed a negative modulation of myofibroblast differentiation and cellular senescence in lung fibroblasts (LFs). These results indicate that the direct administration of bronchial epithelial cell-derived EVs into the trachea may be a potential therapeutic approach to treat IPF by targeting the TGF-β–WNT crosstalk. As described above, another important source of EVs are alveolar epithelial cells. In normal conditions, these structures are characterized by a very slow cellular turnover, but murine models have demonstrated how in the presence of cell damage, AT2 cells acquire significant regenerative properties characterized by their intense cell proliferation and differentiation into AT1 cells. It is well established that EVs represent the way by which AT2 cells communicate with one another in orchestrating the maintenance or repair of damaged alveoli. Quan et al. [188] showed that EXOs derived from A549 cells (an AT2 cell line) could induce the proliferation of AT2 cells via miR-371b-5p, but not differentiation into pluripotent-stem-cell (PSC)-derived cultures. Interestingly, in the same study, it was observed that both bleomycin-treated human-induced PSC (hiPSC)-differentiated AT2 cells (hiPSC-AT2) and primary human AT2 cells were able to secrete miR-371b-5p in EVs. These results are indicative of the potential role of miR-371b-5p as an inducer of re-epithelization in the alveolar niche when lung injury takes place. Another recent study concerning the possible beneficial role of AT2-cell-derived EXOs was carried out by Mitchell and colleagues using a murine model of hyperoxia-induced lung injury [189]. Exosomes were isolated from both hiPSCs and hiPSC-AT2 (alveolar-like phenotype) and bound to magnetic beads before being administrated to mice. It was demonstrated that in the mice that were administered the exosome/beads, the level of tissue injury was lower as compared with that observed in the control mice.

Currently, the roles of EVs derived from mesenchymal cells in epithelial stem/progenitor cells function during regeneration remain unclear. Some studies carried out using mesenchymal cells, such as LFs obtained from lung tissue of patients, have shown that EVs derived from these cells can induce abnormal epithelial regeneration. Haj-Salem et al. have demonstrated that LF-derived EXOs are central in the crosstalk between fibroblasts and epithelial cells [190]. Epithelial cells were challenged with EVs obtained from fibroblasts residing in the bronchi of severe eosinophilic asthma patients, and the effect was that they proliferated abnormally. This hyper-proliferation was due to the lower levels of the antiproliferative mediator TGF-β2.

In a study concerning IPF, Kadota et al. [191] have demonstrated that LF EXOs from patients can be taken up by epithelial cells, causing damage in their mitochondria as well as the induction of senescence. IPF LF EVs induce an overproduction of ROS by the mitochondria, which in turn causes mitochondrial damage, the activation of DNA damage response, and results in senescence. In IPF pathogenesis, it is very important to observe how the interstitial LFs differentiate into contractile myofibroblasts, which are characterized by a high proliferation and an excessive secretion of the extracellular matrix. It is also worth mentioning that some lipidic mediators, such as prostaglandins (PGs), can inhibit myofibroblast differentiation. Some researchers showed that the stimulation of primary human LFs with IL-1β inhibited the differentiation into contractile myofibroblasts of both themselves (autocrine signal) and adjacent naive LFs in co-cultures (paracrine signal). Interestingly, the same study demonstrated for the first time that the effects of the antifibrotic properties of activated fibroblasts on fibroblasts close to them are mediated by EXOs and other EVs, the content of which is represented by several PGs, including the antifibrotic PGE2.

Another important source of EVs in the lungs is represented by resident immune cells with multiple subclasses of dendritic cells, innate lymphoid cells and interstitial and alveolar macrophages. Macrophage-derived EVs can induce regenerative processes in lung epithelia. SOCS1 and SOCS3 (suppressors of cytokine signaling) are two anti-inflammatory mediators contained in EVs that are secreted by alveolar macrophages (AM). They inhibit Janus kinase–STAT signaling, which represents an important pathway activated by many cytokines [192]. Moreover, SOCS3 contained within human alveolar macrophage EVs can exert antitumoral effects and decrease the production of some cytokines involved in allergic response, such as IL-4, IL-13 and thymic stromal lymphopoietin, from airway epithelial cells [193]. Furthermore, in the presence of LPS, macrophages secrete EVs. Moreover, in the early phases of LPS stimulation, these EVs are mostly ABs, which contain miRNAs such as miR-221 and miR-222. Functionally, ABs produced upon the LPS stimulation of macrophages are able to induce the proliferation of malignant and/or normal lung epithelial cells. miR-221/222 deletion in ABs significantly reduces the AB-mediated proliferation, demonstrating that the AB-shuttling of miR-221/222 itself promotes cell growth [194]. Moreover, an important study was carried out with the purpose of evaluating the impact of some miRNAs on the fibrosis associated with IPF progression. Interestingly, the results of this study demonstrated that macrophages isolated from sputum are rich in exosomal miR-142-3p, and that macrophage-derived exosomes inhibit pulmonary fibrosis progression via the delivery of miR-142-3p [195]. It is noteworthy that macrophage EXOs exert an important role as promoters of epithelial cell maintenance.

Resident MSCs represent another important source of EVs in the lung. Several in vivo models have demonstrated the beneficial role of MSC-derived EVs and MVs in inflammatory conditions caused by bacteria and viruses [196,197,198,199]. Silva et al. [200] demonstrated that MSC-EVs were able to improve the alveolar–capillary barrier in experimental models of ARDS (acute respiratory distress syndrome). Even if it is still not clear how dysfunctional mitochondria contribute to the compromised alveolar–capillary barrier, this study demonstrates that MSC-EVs can restore the functionality of mitochondria. The dysfunctional mitochondria respiration, caused by stimulation with LPS of primary human distal lung epithelial cells, is restored by the transfer of mitochondria contained in MSC-EVs. Moreover, in a murine model of an LPS-injured lung, treatment with MSC-EVs reduced lung injury and restored mitochondrial respiration. In another study [201], it was demonstrated how the uptake of MSC-derived EXOs can down-regulate the expression of SAA3 (serum amyloid A3), an important mediator of the acute phase of inflammation. It is noteworthy to specify that this effect has been observed with EVs obtained from miR-30b-3p-overexpressing MSCs. A further demonstration of the anti-inflammatory activity of miRNAs contained within MSC-EVs has been obtained [202,203]. The treatment of murine bone-marrow-derived MSCs with hypoxia increased miR-21-5p concentration in MSC-EXOs, which in turn attenuated ischemia/reperfusion injury in a mouse lung [202].

It has been also demonstrated that autophagy may have an important role in tissue regeneration mediated by MSCs [203,204]. A study by Wei et al. [205] further elaborated on the relationship between autophagy, inflammation and lung damage as exerted by MSC-EXOs. MiR-377-3p contained in EXOs from human umbilical cord MSCs protected lungs from acute injury induced by LPS. This beneficial effect was due to the down-regulation of RPTOR (target regulatory-associated protein of mTOR) mediated by miR-377-3p.

### 2.4. Dysregulation of EVs’ Cargo in the Pathogenesis of COPD and CF

EVs, containing a specific cargo of lipids, proteins, metabolites and nucleic acids, regulate the intercellular communication in both health and disease, influencing physiological and pathological processes (i.e., immune responses [206], tissue repair [207,208], stem cell maintenance [209], cardiovascular diseases [210,211], neurodegeneration and demyelinating diseases [212,213], cancer [214] and inflammation [215]). EXOs can trigger signaling and thus release their content in target cells in different ways: (1) at the cellular level through the interaction of the EXO membrane ligands with specific membrane receptors of recipient cells; (2) by a mechanism in which the membrane proteins of the exosomes are cleaved by the proteases and act as ligands for the target cell receptors; (3) by the direct fusion of the EXO membrane with target cells, resulting in its content being secreted into the cytosol of the recipient cell [216]; (4) at the intracellular level via the process of EXO endocytosis (i.e., clathrin-mediated endocytosis, caveolin-mediated endocytosis, micropinocytosis and phagocytosis) [217] by different types of cells including gastric epithelial cells [218], macrophages [219], cardiomyocytes [220], etc.

EVs, released from almost all cell types in the lung, play an important role in the pathogenesis of chronic respiratory diseases such as COPD and CF [27,221]. They mediate intercellular communication in the lungs and hence are involved in pulmonary homeostasis [222] and cellular response to stimuli and lung disease [221,223]. A dysregulation of the protein, lipid and nucleic acid (especially miRNA) content of exosomes has been highlighted in respiratory diseases that also involve inflammation, especially following exposure to various stimuli [224,225,226]. Several studies have shown that bronchial epithelial cells, which normally regulate airway homeostasis, release more extracellular vesicles than other cells [224]. These EVs are characterized on the surface by some mucins (MUC1, MUC4 and MUC16) [223] that protect against pathogens [227], maintain the structure of the vesicles [223], and allow the interaction of the vesicles with the inhaled substances or with the receiving cells. Epithelial EVs transport pro-inflammatory cytokines and are rich in miRNA-210, whose levels further increase after exposure to cigarette smoke extract (CSE) [228]. Xu et al. showed that CSE increased the levels of miRNA-21 in the EVs released by human bronchial epithelial cells in COPD patients [229]. Moreover, the lengthy exposure to cigarette smoke extract increased cellular communication network factor 1 (flCCN1) levels in EVs, which regulated the lung homeostasis [230]. Alipoor et al. highlighted the importance of airway epithelial cell exposure to tobacco smoke extract in increasing the miRNA content of exosomes such as miRNA-101 and miRNA-144, which inhibit CFTR protein expression in the pathogenesis of COPD [225].

Lung resident macrophage-derived exosomes also play a pivotal role in inflammation, immune function and tissue injury in the lung tissue microenvironment [231]. Macrophage EXOs are enriched with miRNA-223, which is targeted to recipient cells by activating the differentiation of monocytes into macrophages [232], and in MHC II molecules, they are important for antigen presentation and immune activation [233,234]. It has been previously shown that CSE exposition increased the secretion of macrophage MVs with pro-inflammatory activity in lung diseases [235]. Li et al. showed that CSE caused an alteration of the content of macrophage-derived MVs, leading to an up-regulation of MMP14 [162]. Furthermore, Cordazzo et al. showed that mycobacterium infection also increased the protein content of alveolar-macrophage (AM)-derived exosomes, especially HSP-70 [235], with pro-inflammatory activity mediated by NK cell activation [236].

EVs released by endothelial cells, particularly circulating microparticles (EMPs), are increased in patients with several diseases such as COPD [237] or in response to various stresses such as cigarette smoke extract [238]. CSE reduced the trafficking of α1AT mediated by endothelial-cell-derived EVs to lung epithelial cells [239]. Numerous studies showed an increase in CD31+/CD42b- or CD31+CD62E+ EMPs [240,241,242,243] in COPD patients and in healthy smokers, which decreased in smokers who stopped smoking, suggesting the important role of EVs in inducing apoptosis and endothelial damage with poor repair capacity [240]. On the other hand, studies using mouse models investigated the EVs’ cargo following prolonged exposure over time to the cigarette smoke extract, discovering an increase in CD31+/42b- EMPs [244].

EVs also have great potential in CF lung disease. Exosomes derived from CF epithelial cells of the lung airways have a different protein content than the EVs released by healthy cells [245]. EVs mediate inflammation by regulating the migration and activation of neutrophilic leukocytes in the CF airways. This is allowed by the rich content of the CF airway bronchial-epithelial-cell (CFBE41o-)-derived exosomes in the integrin proteins, such as VCAM1 [245], with consequent migration of neutrophils to inflammatory sites [246], and in the ligand S100 A12 [245], which can bind the RAGE receptor placed on the surface of the receiving neutrophils, causing their activation [247].

Differences in EVs’ cargoes between healthy individuals and CF/COPD patients are highlighted in Figure 1.

## 3. EVs and CF

Our research group assessed if the sputum from CF patients contained EVs [248]. These EVs were MPs (100–500 nm) and expressed CD66b (granulocytes, median value of 53.8%), CD11a (leukocytes, median value of 16.1%), and CD11b (monocyte/macrophages, median value of 0%). Thus, according to these data, the MPs were derived from granulocytes (activated or apoptotic), while the presence of monocyte macrophage MPs was not statistically significant. Considering the presence of MPs in the sputum of CF patients, we asked ourselves whether MPs could have a role in the pathophysiology of lung disease. We approached this question by injecting MPs into the lungs of Swiss mice, finding that they caused a massive infiltration of neutrophils in the lung parenchyma and perivascular/peribronchial regions. Interestingly, these neutrophil-derived MPs presented high LPS-like activity, likely as a consequence of the binding of shed LPS in CF airways [249].

A proteomic study on BALF of CF, primary ciliary dyskinesia (PCD) and asthmatic patients identified that PCD and CF respiratory exosomes displayed higher levels of both grancalcin and histones [250], which mediate important properties of neutrophils such as their ability to adhere to fibronectin, their degranulation, and the increase in their killing antimicrobial activity [251]. On the other hand, CF EXOs contained more inflammation-related proteins, such as neutrophil gelatinase-associated lipocalin (LCN2) and S100A12. LCN2 is stored in neutrophil secondary granules and can also be synthesized de novo by epithelial cells and macrophages in response to inflammatory stimuli, acting as a neutrophil chemoattractant [252]. S100A12 is a member of the S100 family of calgranulin small calcium-binding proteins, expressed predominantly in the cytosol of neutrophils and monocytes, and found elevated in CF BALF [253]. It supports neutrophil interactions with fibrinogen, their localization into inflammatory sites, and the expression of cytokines and chemokines via binding to its receptor RAGE (receptor for advanced glycation end products) [254,255]. EXOs collected from the CF patients’ airways were also characterized by high levels of superoxide dismutase (SOD2) and glutathione peroxidase 3 (GPX3), two proteins that can act synergistically in antioxidant defense, and peroxiredoxin 5 (PRDX5), all contributing to the regulation of airway inflammation.

The pathologic role of EVs in CF lung disease was confirmed by Useckaite et al. [245] in an in vitro study. A significantly higher concentration of EVs (50–250 nm, small EV subtype that potentially contains EXOs, small MPs and exomeres) were released from CF airway cell lines (CFBE41o- and CuFi-5) compared with the WT control cell lines (HBE41o- and NuLi-1). CFBE41o- EVs were taken up by healthy donor neutrophils, which increased their CD66 expression, increased myeloperoxidase activity and were stimulated to migrate via the interaction of RAGE on neutrophils and EV-derived S100A12.

More recently, Forrest and colleagues [256] showed that CF sputum EVs can activate naïve neutrophils, inducing both the exocytosis of their primary granule and their concomitant caspase-1 and IL-1β production. Moreover, EVs produced by activated neutrophils allow the delivery of active caspase-1 to primary tracheal epithelial cells which, in turn, activate the inflammasome and release IL-1α, IL-1β, and IL-18. Some studies have also shown the presence of pro-inflammatory EVs, with high protein content, in CF BALF [250] and sputum [249], which are involved in the activation and chemotaxis of neutrophils.

Summarizing all these findings (Figure 2 and Table 1), it has been demonstrated that EVs produced by CFTR-mutated epithelial cells can activate neutrophils and vice versa, that is, neutrophil EVs can activate epithelial cells, namely their inflammasome (Figure 2). What has been a conundrum of CF lung disease, i.e., how a chronic disease is characterized by a neutrophil-dominated inflammation, a typical acute inflammatory cell type, can be partly explained by the secretion of two neutrophil-activating proteins (LCN2 and S100A12) at a distance from the initial lesion by EVs. Indeed, EVs produced in discrete areas of the bronchiolar tree may well induce a spreading of signaling (involving inflammasome) via uptake by resident, long-lived epithelial cells, but also in freshly recruited neutrophils.

## 4. EVs and COPD

Fujita and colleagues [228] investigated whether CSE modified epithelial EVs, thus considering their role as novel paracrine intercellular communication mediators during airway remodeling in COPD pathology. LFs took up EVs (50 to 150 nm, i.e., in the EXO range) secreted by immortalized BEAS-2B cells or primary HBECs, with EVs obtained from cigarette smoke extract (CSE)-treated HBECs promoting the expression of fibrotic markers in LFs, that is, collagen type I and α-smooth muscle actin (α-SMA). miR-210 was enriched in HBEC-derived EVs and promoted myofibroblast differentiation in LFs through the inhibition of autophagy protein ATG7. Thus, the inhibition of autophagy mediated by transferred EVs allowed the myofibroblast to differentiate in response to tobacco smoke. This process seems to be fundamental in airway remodeling in COPD pathogenesis.

In vitro and in vivo experiments have demonstrated that another miRNA, miR-21, is involved in bronchial epithelial cell-derived exosome (≤200 nm)-mediated airway remodeling [229]. The levels of miR-21 were higher when MRC-5 cells (lung fetal fibroblasts) were co-cultured with CSE-treated HBE cells, but not in the co-culture of MRC-5 cells and normal HBE cells. When MRC-5 cells were in the same co-culture conditions, an up-regulation of collagen I and α-SMA was observed. The transfection of HBE cells with an miR-21 inhibitor and the subsequent treatment with CSE did not exert any effect on α-SMA or collagen I expression. On the other hand, both the treatment of co-cultures with GW4869, an inhibitor of exosome generation, and the down-regulation of exosomal miR-21 in HBE cells had an inhibitory effect on the differentiation of myofibroblasts. To gain insight into the signaling pathway leading to fibroblast differentiation caused by exosomal miR-21, the role of HIF-1α, a transcription factor involved in fibrosis and myofibroblast differentiation [257], was investigated. The application of EXOs produced by CSE-treated HBE cells onto MRC-5 cells induced the up-regulation of HIF-1α. Consistently, the levels of pVHL, a factor related to HIF-1α stabilization, were decreased. The gene silencing of HIF-1α reduced the levels of α-SMA and collagen I that were induced by CSE-treated HBE cell EXOs. When CSE-treated HBE EXOs were applied to MRC-5 cells, it was possible to observe a change in the expression of pVHL, HIF-1α, α-SMA, and collagen I. Interestingly, the down-regulation of exosomal miR-21 blocked these changes in expression, but the gene silencing of pVHL decreased the effect of the miR-21 inhibitor. Finally, in a BALB/c mouse model of COPD obtained with an eight-week exposure to CS, when mice were injected with an antagomir-21 via the tail vein weekly, simultaneously with the first CS exposure in CS-exposed mice, the down-expression of miR-21 prevented changes in pulmonary function and attenuated inflammation and airway obstruction. Moreover, the expression of pVHL was improved when mice were treated with antagomir-21, whereas a contraction of HIF-1α levels was observed. Overall, these results highlight that CS can modify the cargo of exosomes and identify miR-21 produced in the bronchial epithelial cells as a key factor for the differentiation of myofibroblasts through the pVHL/HIF-1α signaling pathway. However, the results revealed that the levels of α-SMA and collagen I were only partly regulated by miR-21/pVH, probably as a consequence of the regulation of airway fibroblast differentiation by multiple pathways. Indeed, the TGF-β1/WNT and the PI3K/AKT pathways are involved in fibroblast differentiation and the up-regulation of collagen I/III in COPD, respectively [258,259]. It could be interesting to see whether exosomal miR-21 can also mediate myofibroblast differentiation through these other pathways. Furthermore, the role of exosomes in the dysfunctional crosstalk of the various cell types involved in airway remodeling (goblet cells, smooth muscles cells, and lung fibroblasts) needs to be clarified.

Emphysema is due to the destruction of alveolar tissues due to neutrophil influx into the airways and elastin degradation by a protease overwhelming of antiproteases, mainly α1-AT, and epithelial apoptosis. To study these pathological features, Moon and colleagues focused on the role of CCN1, also called Cyr61, a cysteine-rich, 38 kD secreted protein that is expressed by several cytotypes comprising airway epithelial cells and that induces IL-8 secretion by these cells [260]. There existed two forms of CCN1, including a full-length (fl)CCN1 that was cleaved into cCCN1 and found in the extracellular matrix [230]. However, the secretion pathway of the two forms differed: while the larger part of flCCN1 was identified in exosomes (as verified by size range (TEM) and Rab27a expression), cCCN1 was found in the supernatant of the conditioned medium as a soluble component. Secreted plasmin (upon CSE challenge) was responsible for the flCCN1 cleavage at the extracellular level. Only exosomal flCCN1was involved in the secretion of both IL-8 and VEGF by airway epithelial cells, while the soluble fraction (i.e., cCCN1) was mainly implicated in the secretion of MMP1. Integrins, particularly α_7_ chains, were implicated in the cCCN1-mediated MMP1 secretion. By using a mouse emphysema model, by which emphysematous changes occurred after six months of exposure to CS (but not at one month), it was observed that the BALF cCCN1 level was increased in samples collected from mice after six months of CS exposure but not from those after one month, showing that CCN1 levels correlated with the development of emphysema [230]. Overall, these data suggest that surface exosomal flCCN1 is cleaved by plasmin and cCCN1 interacts with integrin-α_7_ and activates the secretion of MMP1 in lung epithelial cells. A complex role for flCCN1 is also suggested by this study. flCCN1 may be well suitable to be a fundamental signaling molecule in lung homeostasis and pathophysiology, since it induces inflammatory responses via mediating IL-8 secretion and subsequent neutrophil enrollment, whereas, on the other hand, it may also have a role in sustaining the integrity of lung tissue by facilitating VEGF secretion. Interestingly, the progressive increase in cCNN1 during CS exposure induced MMP1 secretion while decreasing that of VEGF [230], implying a role for CCN1 in elevating MMP1 and down-regulating VEGF, alterations which have been well documented in inducing emphysema [261,262,263]. As concerning exosomes, another implication of this study is that the exosome-mediated secretion of flCCN1 potentially allows the propagation of “inflammatory” signals to distant portions of the lungs.

Besides epithelial cells, other cell types have been implicated in EV release and the pathophysiology of COPD, such as macrophages and endothelial cells. Macrophages are involved in emphysematous lung destruction [264,265] through their expression of a variety of proteases [264,266,267]. Li and colleagues [162] found that tobacco smoke extract induced macrophages to release MVs (isolated by ultracentrifugation at 100,000× *g*) which possess proteolytic activities owing to a single transmembrane MMP, namely MMP14, which is also known as the membrane type 1 MMP. Based on biochemical and morphological studies on THP-1 monocytes differentiated into macrophages and primary human monocyte-derived macrophages, it was demonstrated that the release of these MVs (approximately 1 μm by confocal microscopy) depends on dynamic, regulated steps that include the activation of the JNK and p38 MAPKs, the induction of cellular MMP14 via MAPK, the cleavage of pro-MMP14 into its active mature form, an increase in MMP14 into shedding plasma membrane blebs, and finally, caspase- and MAPK-dependent apoptosis and apoptotic blebbing. Further studies to explore the role of macrophage MVs in the damage of the lung extracellular matrix in COPD are warranted.

MPs were identified in the sputum of COPD patients and were released by different cell types, such endothelial cells, granulocytes, monocytes/macrophages, platelets/megakaryocytes, and red blood cells by flow cytometric analysis [268]. Notably, those derived from endothelial cells negatively correlated with forced expiratory volume in 1 s (FEV_1_), likely reflecting the increased apoptosis of endothelial cells [269] with the worsening of the clinical condition. The possible role of endothelial-cell-derived MPs was investigated by Lockett et al. [239], who searched for α1AT release in the lung across an intact pulmonary endothelial barrier. In primary rat pulmonary endothelial cell monolayers, it was shown that a nonclassical pathway of α1AT secretion was effectuated via the release of MPs (isolated at 100,000× *g*). Interestingly, CSE exposure reduced the levels of trans-cytosed α1AT, suggesting that CS may inhibit α1AT delivery into epithelial cells mediated by EVs secreted by endothelial cells, this process being fundamental in COPD pathogenesis.

More recently, a role for neutrophil EVs in emphysema was disclosed by Genschmer et al. [270] and Margaroli and colleagues [271]. In the first of these studies [270], it was found that EXOs (~100 nm by NTA) from activated neutrophils were enriched with surface neutrophil elastase (NE), which was enzymatically active and resistant to inhibition by α1AT. When exosomes were treated with purified human NE, their configuration changed, exposing the NE activity on their surface, implying that the loading of exosomes with NE occurs during the exocytosis of primary granules in the close proximity of released exosomes. Activated, but not quiescent, exosomes visibly destroyed the collagen fibrils via the integrin Mac-1 αM-I domain and bound to elastin fibers. In a model of murine COPD by intratracheal injection, exosomes from activated, but not quiescent, neutrophils caused alveolar enlargement, increased airway resistance, and right ventricular hypertrophy (RVH), compared to control mice. To determine the collagen disruption, the murine BALF was then measured for the quantity of acetyl-proline-glycine-proline (PGP), a product derived from collagen degradation, associated with ECM turnover and generally increased in lung secretions from COPD patient [272]. The PGP levels in the BALF of the activated-exosome-treated but not quiescent-exosome-treated animals were significantly increased. In mice treated with BALF CD66+ exosomes collected from COPD patients (~120 nm by NTA), it was possible to observe a marked alveolar enlargement and RVH. On the contrary, mice administered with pooled BALF exosomes from healthy controls did not present pathological changes. The majority of CD66b+ exosomes (96.1%) in COPD BALF expressed NE, whereas few (1.3%) of those collected from healthy, nonsmoking (NS) control BALF expressed NE, suggesting that COPD BALF exosomes principally induce alveolar enlargement by NE present on the CD66b+ population of exosomes (whereas NS CD66b+ purified BALF exosomes did not).

Margaroli et al. [271] extended these findings by implementing a mouse-to-mouse EV transfer model. Intratracheal administration of LPS in A/J mice allowed to determine the presence of proteolytically active α1AT-resistant neutrophil exosomal NE in BALF 24 h later. The administration of these EVs (EXO size range of 50–100 nm) in a single dose to A/J or C57BL/6 mice resulted in alveolar enlargement and a dose-response relationship, with significant effects observed with 1 × 10^6^ EVs and a robust response beginning at 1 × 10^7^ EVs. EV-mediated alveolar enlargement from a single 1 × 10^7^ EV dose lasted for at least three weeks, with the induction of low levels of inflammation at weeks one and two. The alveolar enlargement translated into local and systemic pathological changes, in that mice showed increased airway resistance, lower forced expiratory volume in 0.1 s (FEV0.1), and right ventricular hypertrophy one week after dosing. When EVs were isolated from NE-KO (Elane^–/–^) mice after the induction of airway inflammation via LPS, the alveolar enlargement effect was lost. The same result was obtained when the EV neutrophil-derived population was depleted via a bead-based pulldown method (Ly6G^+^).

Summarizing all of these studies, a complex role of EVs in COPD pathogenesis emerges (Figure 3 and Table 1). Many cell types are involved, such as airway and respiratory epithelial cells, lung macrophages, endothelial cells and neutrophils. It is likely that EVs participate in many amplifying steps in COPD pathogenesis, including inflammation, tissue destruction and fibrosis.

**Table 1 ijms-24-00228-t001:** Pathogenic roles of EVs in CF and COPD.

Study	Ev Type	Lung Disease	Study Type	Pathogenic Outcomes
Porro et al., 2010 [248]	Microparticles (100–500 nm)	Cystic fibrosis	Patients	MPs were found in CF sputa and were mostly of granulocyte origin (CD66b^+^), while leukocytes (CD11a^+^) and monocyte/macrophages (CD11b^+^) MPs were less present.
Porro et al., 2013 [249]	Microparticles (100–500 nm)	Cystic fibrosis	In vivo, in mice	I.t. injection of sputum MPs obtained from a CF patient in acute conditions in Swiss mice induced peribronchial/perivascular infiltrates, similar to that obtained with LPS, whereas the inflammatory response was lower when MPs obtained from stable CF patients were injected.
Rollet-Cohen et al., 2018 [250]	Exosomes (20–150 nm)	Cystic fibrosis	In vivo, in patients	BALF exosomes CF patients were enriched with proteins implicated in neutrophil function, such as chemotaxis (LCN2, S100A12) and degranulation (grancalcin), antioxidant proteins (SOD2, GPX3, PRDX5), antiproteases (SERPINAA6), and those involved in the response to the chronic infectious challenge (histones, TOLLIP).
Useckaite et al., 2020 [245]	50–250 nm (exosomes, small microparticles and exomeres)	Cystic fibrosis	In vitro	Higher EV levels were found in CFBE41o- and CuFi-5 cells as well as in BALF of patients with CF. A significant increase in neutrophil chemotaxis was observed with CFBE41o-EVs as compared to control EVs.
Forrest et al., 2022 [256]	<400 nm	Cystic fibrosis	Ex vivo	EVs isolated from CF sputa were positive for active caspase 1, induced transmigration of neutrophils, which showed primary granule exocytosis and increase in intracellular active caspase-1. Neutrophil-derived EVs up-regulated caspase 1 in primary tracheal cells.
Li et al., 2013 [162]	Microparticles	COPD (emphysema)	In vitro	MVs released from TSE-exposed macrophages (THP-1 macrophages and hMDMs) carried substantial gelatinolytic and collagenolytic activities that could be predominantly attributed to a transmembrane MMP14. TSE induced MMP14 accumulation in small, circumscribed cell-surface domains and an increase in MMP14-enriched MPs. Activation of the JNK and p38 MAPKs and apoptosis were a requisite for MMP14-positive MVs release.
Moon et al., 2014 [230]	Exosomes (by TEM)	COPD (emphysema)	In vitro, in vivo	CSE increased the percentage of CCN1-positive exosome in bronchial Beas2B cells. IL-8 and VEGF secretion was increased by exosomal full-length (fl)CCN1. MMP-1 secretion was increased predominantly by secreted cleaved (c)CCN1. In C57BL/6 mice exposed to CS, emphysematous changes were observed after 6 months. The cCCN1 level in BALF was highly elevated in BALF obtained from mice after 6-month CS exposure but not from those after 1-month exposure.
Fujita et al., 2015 [228]	50–150 nm (exosomes)	COPD	In vitro	EVs derived from cigarette smoke extract-stimulated bronchial epithelial cells promoted lung fibroblasts to differentiate into myofibroblasts through miR-210 transfer and autophagy suppression.
Lacedonia et al., 2016 [268]	MPs	COPD	Patients	MPs were isolated from the sputum of mild to severe COPD patients and were positive for CD66b (granulocytes), CD235ab (erythrocytes), CD31 (platelets/endothelial cell adhesion molecules 1), CD41 (platelets/megakaryocytes), and CD11a (leukocytes). There was a negative correlation between CD31-MPs and FEV_1_, whereas CD66b-MPs were correlated with a worse COPD performance index.
Xu et al., 2018 [229]	≤200 nm (exosomes)	COPD	In vitro and in vivo	Higher levels of α-SMA and collagen 1 in MRC-5 cells (bronchial fibroblasts) exposed to exosomes derived from CSE-treated HBE cells. In MRC-5 cells, down-regulation of exosomal miR-21 blocked the exosome-induced myofibroblastdifferentiation phenotype as well as the increased and decrease levels of HIF-1α and pVHL respectively. In BLAB/c mice exposed to CS for 8 weeks, down-expression of miR-21 prevented changes in pulmonary function and attenuated inflammation and airway obstruction, restored pVHL expression and decreased HIF-1α levels.
Genschmer et al., 2019 [270]	~100 nm (exosomes)	COPD	In vitro, in vivo	Exosomes from activated neutrophils had considerably higher quantities of surface NE compared to quiescent exosomes. Exosomal NE was resistant to inhibition by α1AT. Activated, but not quiescent, exosomes destroyed the collagen fibrils over time. When exosomes were administered intratracheally into A/J mouse airways, activated but not quiescent neutrophil exosomes caused the hallmarks of COPD, alveolar enlargement, increased airway resistance, and RVH, compared to mice treated with PBS. Human COPD lung-derived CD63^+^/CD66b^+^ exosomes induced marked alveolar enlargement and RVH.
Margaroli et al., 2022 [271]	50–100 nm (exosomes)	COPD (emphysema)	In vivo	Higher levels of BALF neutrophil-derived exosomes in LPS-treated mice (A/J) than controls. Dose-dependent alveolar enlargement upon i.t. administration of LPS-derived exosomes in A/J and C57BL/6 mice.

α1AT: α1-antitrypsin; BALF: bronchoalveolar lavage fluid; CF: cystic fibrosis; COPD: chronic obstructive pulmonary disease; CS: cigarette smoke; CSE: cigarette smoke extract; EVs: extracellular vesicles; FEV_1_: forced expiratory volume in 1 s; hMDMs: human monocyte-derived macrophages; i.t.: intra-tracheal; MMP: metalloproteinase; MPs: microparticles; NE: neutrophil elastase; ROS: reactive oxygen species; RVH: right ventricular hypertrophy; TSE: tobacco smoke extract.

## 5. Role of EVs as Biomarkers in CF and COPD

The cargoes transported by EVs are cell-specific and can allow the use of EVs as diagnostic tools for a variety of pathologies, including cerebrovascular disease [273,274], diseases of the central nervous system [275], kidney [276], liver [277], lungs [16,278], and cancer [113,279,280,281,282]. The EV/EXO enrichment in several different biospecimens (serum/plasma, BALF, urine, cerebrospinal fluid, saliva, milk or other fluids), and the observations that their composition can discriminate a disease state vs. normal conditions, emphasize the role of EVs as novel circulating biomarkers [283]. Importantly, besides the fact that the physiological state and microenvironment of their cells of origin are reflected by EV subtypes, and that most cells secrete EVs with specific cargoes [284,285,286], their membrane protects internal contents, which are transferred from the parent cell cytosol to recipient cells after EV–cell fusion occurs. Although, in principle, the EVs’ phospholipid bilayer protects their content and makes them very stable in the extracellular environment, exogenously administered EVs show a relatively short half-life in circulation (in the range of tens of minutes), implying that EVs to be considered as reliable biomarkers should be dosed serially or differentially [287]. However, they possess long shelf stability [104].

Recent evidence, as explained above, points to the pathogenic role of EVs in CF and other neutrophilic airway diseases. EVs have been found in sputum and BALF samples obtained from people with CF, a finding compatible with the higher EV secretion from CF bronchial epithelial cells [245]. Although many inflammatory and remodeling biomarkers have been found and exploited in CF [288,289,290], only few of them have reached a clinical application, such as serum high sensitivity C-reactive protein and sputum neutrophil elastase [291]. Proteomic analyses have found many putative biomarkers in CF BALF, which may be involved in pathogenesis of lung disease. Rollet-Cohen and colleagues [250] detected 14 proteins that were differentially abundant amongst CF, asthma and PCD. A recent study highlighted that urine EXOs and their cargoes might be employed as noninvasive biomarkers, since these EXOs showed a clear separation in differentially expressed proteins between CF patients and healthy controls [292].

Another biomarker found in airway EXOs is prolyl endopeptidase (PE), an extracellular protease originating from collagen, which is a neutrophil chemoattractant [293] implied in the pathogenesis of neutrophilic chronic lung diseases [294,295]. Increased PE-EXO release was observed in airway epithelial cells via a TLR4-mediated mechanism, and the presence of protease-rich EXOs in the sputum specimens obtained from subjects with CF lung disease colonized by Gram-negative bacteria *P. aeruginosa* was determined [296]. Thus, PE-EXO may represent another biomarker of CF lung diseases.

However, this insightful exploration of EV cargoes in airway-derived fluid and urine has not translated to any biomarker exploitation for CF.

Altered protein and RNA profiles in COPD have been discovered by studying blood EVs [224]. Proteomic studies highlighted differentially expressed proteins in smokers, patients with COPD and nonsmokers. Sundar et al. [297] were the first to find that several plasma proteins (CD5 antigen-like, fibronectin, clusterin, gelsolin, hyaluronan-binding protein, apolipoprotein D, and EGF-containing fibulin-like ECM protein) were differentially enriched in smokers and patients with COPD compared to nonsmokers.

The study of circulating EMPs has been considered in the context of COPD disease pathomechanisms and biomarker discovery. Elevated levels of EMPs have been found in subjects with COPD [298,299,300,301]. Since not all those who smoke develop COPD, EMP quantitation and cargo analysis may have relevance for considering subjects at risk.

Sundar et al. [302] were also the first to explore plasma-derived EV small RNA cargoes from nonsmokers, smokers, and patients with COPD. EVs from nonsmokers (n = 6), smokers (n = 6) and patients with COPD (n = 8) were similar to each other both in the size distribution of EXOs and the total number of particles. These EVs were shown by RNA-seq analysis to be enriched with miRNAs, tRNAs, piRNAs, snRNAs, snoRNAs. Focusing on miRNAs as novel biomarkers in smokers and patients with COPD for further analysis, they found distinct miRNA profiles (up-regulated: miR-22-3p, miR-99a-5p, miR-151a-5p, miR-320b, miR-320d; and down-regulated: miR-335-5p, miR-628-3p, miR-887-5p and miR-937-3p) in COPD versus smokers or nonsmokers. Interestingly, a gene ontology analysis of nonsmokers vs. COPD comparison showed an enrichment of smoking-mediated pathways such as apoptosis, inflammation-related signaling, and oxidative stress response.

COPD patients experience a brisk reduction in airflow during acute exacerbation episodes, which are due to bacterial and/or viral infections, while specific causes could not be identified in about 30% of cases. Since EMPs may be involved in COPD pathogenesis, Takahashi and colleagues compared the levels of CD144+ MPs (VE-cadherin EMPs), CD31+/CD41- MPs (PECAM MPs), CD146 MPs (MCAM MPs) and CD62E+ MPs (E-selectin MPs) in stable COPD patients, patients with exacerbated COPD, and healthy non-COPD individuals [299]. It was found that VE-cadherin, PECAM and E-selectin EMP numbers were significantly higher in the patients with exacerbated COPD than in the stable COPD patients. Moreover, E-selectin EMP levels were significantly higher in COPD patients with frequent exacerbations than in those without (*p* < 0.001), and, interestingly, returned to levels similar to those of stable COPD patients without a history of frequent exacerbation on day 28, while the clinical exacerbation lasted until 14 days. A significant outcome of this study is that high E-selectin EMPs in patients with frequent exacerbations may predict COPD patients who may be susceptible to exacerbation.

In another study, circulating CD9+ EVs (i.e., exosomes) were found to be increased in patients with acute exacerbation vs. stable COPD patients vs. controls [303]. These EV levels correlated with systemic inflammatory markers (CRP, sTNFR1, IL-6 in plasma).

It can be concluded that plasma EVs may mediate important processes during inflammation in COPD-associated acute exacerbations that require further investigation.

## 6. Conclusions

The exploration of EVs of different sources in CF and COPD has determined that many of the mechanisms involved in pathobiology are mediated by MPs and EXOs. A piece of the puzzle which is missing is if the system is redundant or specific clues may be obtained from this knowledge. An insightful consideration about these issues may bring novel therapies which, in a favorable way, could lead to personalized medicines. For example, we do not know whether CF patients bearing a different class of mutations have EVs that behave in the same manner. Compounding with these studies, an effort towards the standardization of the isolation, characterization and functional assays for EVs isolated from airway cells should be brought forth. Finally, there is still room for understanding the richness of EVs in biological fluids (e.g., BALF, sputum) and their relevance as biomarkers. Prospective studies are necessary for the comprehension of EVs as theranostic (diagnostic and therapeutic) and prognostic biomarkers.

## Figures and Tables

**Figure 1 ijms-24-00228-f001:**
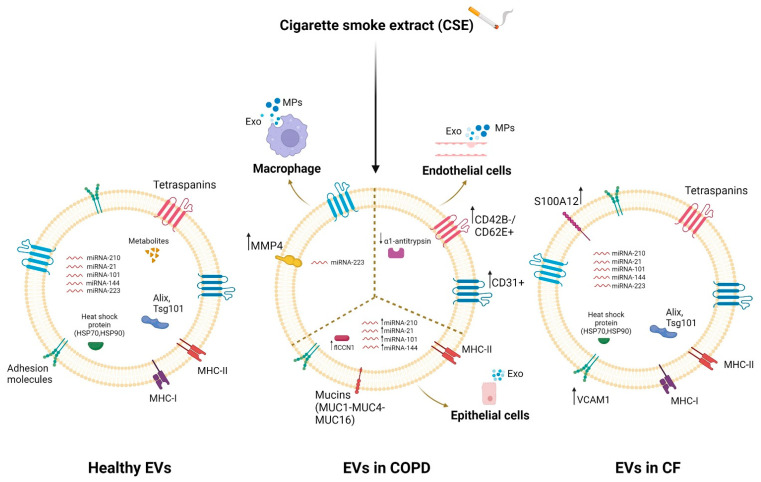
Molecular cargo of healthy EVs and vesicles released by various cell types in the pathogenesis of chronic obstructive pulmonary disease (EVs in COPD) and cystic fibrosis (EVs in CF). The increase in some vesicle’s components, involved in certain respiratory diseases, is indicated by an upward arrow. The exposure to various stimuli such as cigarette smoke extract leads to changes in the content of EVs released by the various cell types in the lung. In COPD, MMP4 and miRNA-223 levels increase in the vesicles released by macrophages; epithelial-cell-derived EVs show an overexpression of miRNA-101, miRNA-21, miRNA-210, miRNA-144 and an increase in the cellular communication network factor 1 (flCCN1) levels; endothelial cells secrete exosomes and microparticles (MPs) rich in CD31+/CD42b- or CD31+/CD62E+ and with reduced levels of α1-antitrypsin. In CF, the expression of the integrin protein VCAM1 and the ligand S100A12 is enhanced in the EVs released by the airway bronchial epithelial cells. Created with BioRender.com.

**Figure 2 ijms-24-00228-f002:**
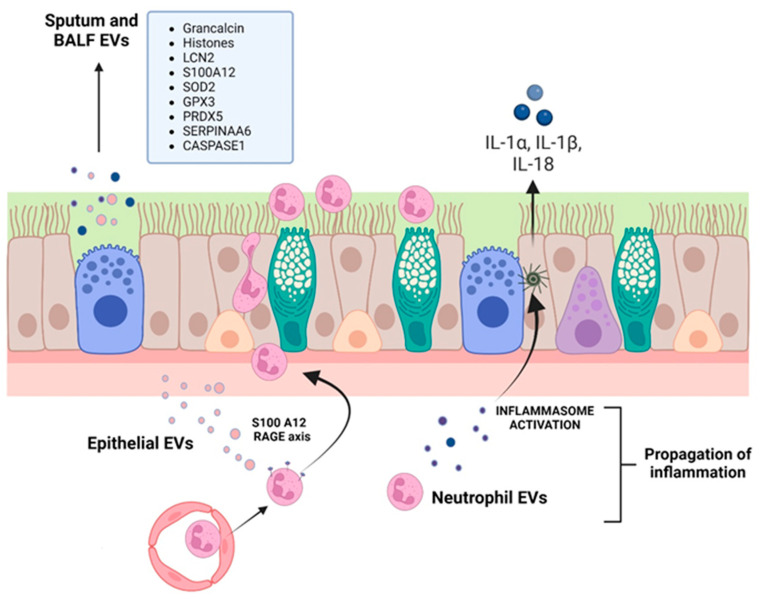
Involvement of EVs in CF inflammatory lung disease. EVs released by airway epithelial cells and enriched with S100A12 increase RAGE on blood neutrophils, which transmigrate and exudate into the CF airway lumen. On the other hand, neutrophil-derived EVs induce inflammasome activation in airway epithelial cells and the subsequent secretion of the cytokines Il-1α, IL-β and IL-18, thus propagating inflammation. All these EVs are then transported into the airway lumen and found in CF airway secretions (sputum, BALF). Created with BioRender.com.

**Figure 3 ijms-24-00228-f003:**
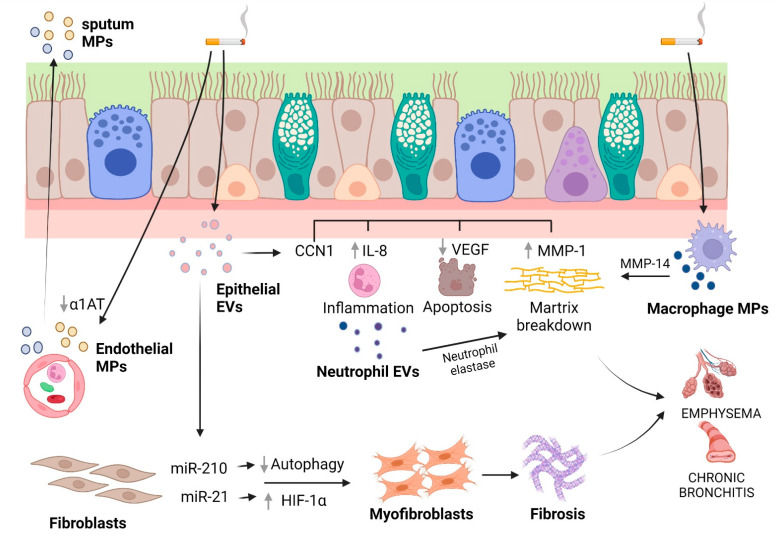
Involvement of EVs in COPD inflammatory lung disease and remodeling. Cigarette smoke induces EV release, which due to their miRNAs cargo cause reduced autophagy (miR210) or an increase in HIF-1a, both events leading to fibroblast differentiation in myofibroblasts and eventually lung fibrosis. Epithelial EVs induced by cigarette smoke are enriched with CCN1, which has various effects, such as IL-8 secretion (and neutrophil recruitment), decreased VEGF (and apoptosis), and increased MMP1 levels (and matrix breakdown). Matrix degradation is also due to macrophage MPs, whose release is triggered by smoke, and that are enriched with MMP-14. Endothelial-cell-derived MPs are enriched with α1AT, which could be down-regulated by smoke. They have been found also in the sputum of CF patients, together with those derived by circulating granulocytes, platelets, and red blood cells. Created with BioRender.com.

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
