# Peer review of "Extracellular Vesicles’ Role in the Pathophysiology and as Biomarkers in Cystic Fibrosis and COPD"

_ijms, 2022, doi:10.3390/ijms24010228_

Round 1

Reviewer 1 Report

In the present review, Authors explore the role of extracellular vesicles in the lung homeostasis and in the pathogenesis of cystic fibrosis and COPD. The knowledge about EV function in the pathophysiology of lung disease is still incomplete, but emerging evidence suggest a pivotal role of EVs that might even fucntion as biomarkers.

The Review is complete and provides an exhaustive overview of the field that will be hopefully further explored in the near future.

Here below my major concerns:

-  The classification of EVs in MV EXO and Apoptotic bodies is correct, but reductive, considering that significant steps forward have been done by the EV community in this research field, discovering a huge variety of EVs so far. Considering this observation, I suggest to remove thestatement "namely apoptotic bodies, microvesicles and exosomes" in the second row of the abstract.

- I suggest to substitute - whenever possible - the MV end EXO nomenclature with the more appropriate EV wording. This is advisable whenever the biogenesis of the obeserved vesicle is uncertain or unexplored. For example, line 158 and 169, but the same revision should be applied to the entire manuscript.

- I suggest to remove the section 2.4, because - in my opinion - it deals with a complex issue in a superficial and too synthetic manner. I guess it is not the focus of this review to deal with technical issues, thus it might be helpful to sinthetically cite the complexity of the isolation and characterization steps in the previous sections, referring simply to specific reviews and position papers of the ISEV task forces.

- Section 2.5 deals with interesting data on the role of EVs in the pathogenesis of COPD and CF, but this section seems to be redudant and several studies cited here are further described in sections 3 and 4. I suggets to include the significant statements of section 2.5 in the corresponding disease specfic sections (i.e 3 or 4) and remove section 2.5

- Minor: too many abbreviations that make difficult to read some of the sections. Are they really necessary? Examples: LSE, DC, CSF, etc seem to be cited only once.

Author Response

In the present review, Authors explore the role of extracellular vesicles in the lung homeostasis and in the pathogenesis of cystic fibrosis and COPD. The knowledge about EV function in the pathophysiology of lung disease is still incomplete, but emerging evidence suggest a pivotal role of EVs that might even fucntion as biomarkers.

The Review is complete and provides an exhaustive overview of the field that will be hopefully further explored in the near future.

Here below my major concerns:

Q: The classification of EVs in MV EXO and Apoptotic bodies is correct, but reductive, considering that significant steps forward have been done by the EV community in this research field, discovering a huge variety of EVs so far. Considering this observation, I suggest to remove thestatement "namely apoptotic bodies, microvesicles and exosomes" in the second row of the abstract.

A: We have removed the statement as indicated by the Reviewer. Nevertheless, we would draw the attention of the Reviewer that most studies on CF and COPD cited in the manuscript report the isolation and characterization of EXOs/MVs, thereby the meaning of the statement that we have inserted in the Abstract.

Q: I suggest to substitute - whenever possible - the MV end EXO nomenclature with the more appropriate EV wording. This is advisable whenever the biogenesis of the obeserved vesicle is uncertain or unexplored. For example, line 158 and 169, but the same revision should be applied to the entire manuscript.

A: To cope with the Reviewer’s concern, we have introduced in Section 2.1 the issue of EV heterogeneity and, accordingly, we have indicated, whenever it was possible, the more appropriate wording. We hope that our effort will meet the Reviewer’s concerns.

Q: I suggest to remove the section 2.4, because - in my opinion - it deals with a complex issue in a superficial and too synthetic manner. I guess it is not the focus of this review to deal with technical issues, thus it might be helpful to sinthetically cite the complexity of the isolation and characterization steps in the previous sections, referring simply to specific reviews and position papers of the ISEV task forces.

A: Section 2.4 was removed and EVs isolation/characterization issues were succinctly proposed in previous sections.

Q: Section 2.5 deals with interesting data on the role of EVs in the pathogenesis of COPD and CF, but this section seems to be redudant and several studies cited here are further described in sections 3 and 4. I suggets to include the significant statements of section 2.5 in the corresponding disease specfic sections (i.e 3 or 4) and remove section 2.5

A: The aim of Section 2.5 is to give an account on the modifications of EV cargo in CF and COPD as compared to healthy condition, since it is known that cargoes amount and identity change in pathological states. The Figure 1 was intended to represent this Section. For these reasons, we would like to maintain the Section, although, accordingly with the Reviewer’ suggestion, we have removed all sentences overlapping with Sections 3 and 4, which were meant to define the EV pathophysiologic role in CF and COPD.

Q: Minor: too many abbreviations that make difficult to read some of the sections. Are they really necessary? Examples: LSE, DC, CSF, etc seem to be cited only once.

A: We have done our best to avoid the unncessary abbreviations.

Reviewer 2 Report

Specific Comments:

-In the legend of figure 3, the authors mention the platelets but they are not illustrated in figure 3.

-Page 3, the paragraph EVs and lung diseases: it would be relevant to also discuss the work that has shown the involvement of EVs in pulmonary embolisms occurring in COVID-19. Here are some relevant references:

PMID: 32944185

PMID: 32622963

PMID: 32938299

PMID: 35929616

-The intro is too short. Authors should further develop the general role of EVs by citing and discussing other references:

PMID: 28684288

PMID: 30184457

PMID: 34067503

PMID: 34182130

PMID: 35752228

PMID: 36174608

Author Response

Specific Comments:

Q: In the legend of figure 3, the authors mention the platelets but they are not illustrated in figure 3.

A: We now present a revised Figure 3 depicting EVs released by circulating plateletes, red blood cells and granulocytes.

Q: Page 3, the paragraph EVs and lung diseases: it would be relevant to also discuss the work that has shown the involvement of EVs in pulmonary embolisms occurring in COVID-19. Here are some relevant references:

PMID: 32944185

PMID: 32622963

PMID: 32938299

PMID: 35929616

A: We have now introduced this issue and relevant references in the Introduction Section.

Q: The intro is too short. Authors should further develop the general role of EVs by citing and discussing other references:

PMID: 28684288

PMID: 30184457

PMID: 34067503

PMID: 34182130

PMID: 35752228

PMID: 36174608

A: We have now extended the Introduction presenting and discussing the references indicated by the Reviewer (citing also other references).

Reviewer 3 Report

Summary:

In this review manuscript, the authors described update knowledge of the extracellular vesicles, apoptotic bodies, microvesicles and exosomes and their role in cystic fibrosis and chronic obstructive pulmonary disease.

General concerns:

1.     Line 124: Please explain “a.k.a.” in ‘Microvesicles (MVs), a.k.a. ectosomes, are smaller than apoptotic bodies’.

2.     Lines 181-182: Please correct “(I, II, V e VI)” in “Furthermore, EXOs carry other proteins like annexins (I, II, V e VI) and Ras proteins that promote the docking of the---”.

3.     Abbreviations: Define abbreviations upon first appearance in the text. Line 443: “EVs released by endothelial cells, particularly circulating microparticles (EMPs)”. Line 755: “The study of circulating endothelial microparticles (EMPs) has been considered---”. Please delete one EMPs.

4.     Line 524: Please correct ‘vice versa’ in “CFTR-mutated epithelial cells were able to activate neutrophils and viceversa”. Please correct all the typos throughout the manuscript.

Author Response

In this review manuscript, the authors described update knowledge of the extracellular vesicles, apoptotic bodies, microvesicles and exosomes and their role in cystic fibrosis and chronic obstructive pulmonary disease.

General concerns:

  1. Line 124: Please explain “a.k.a.” in ‘Microvesicles (MVs), a.k.a. ectosomes, are smaller than apoptotic bodies’.

“a.k.a.” stays for “also known as”. We have corrected the manuscript accordingly.

  1. Lines 181-182: Please correct “(I, II, V e VI)” in “Furthermore, EXOs carry other proteins like annexins (I, II, V e VI) and Ras proteins that promote the docking of the---”.

Sorry, but we do not understand which correction should be introduced.

  1. Abbreviations: Define abbreviations upon first appearance in the text. Line 443: “EVs released by endothelial cells, particularly circulating microparticles (EMPs)”. Line 755: “The study of circulating endothelial microparticles (EMPs) has been considered---”. Please delete one EMPs.

We have deleted the second “EMPs”. In general, we have tried to define all abbreviations upon first appearance.

  1. Line 524: Please correct ‘vice versa’ in “CFTR-mutated epithelial cells were able to activate neutrophils and viceversa”. Please correct all the typos throughout the manuscript.

We have corrected this typo, and in general we have done our best to correct all typos.